# An Analysis of Polysaccharides from Eight Plants by a Novel Heart-Cutting Two-Dimensional Liquid Chromatography Method

**DOI:** 10.3390/foods13081173

**Published:** 2024-04-12

**Authors:** Haonan Wang, Hongyu Jin, Ruiping Chai, Hailiang Li, Jing Fan, Ying Wang, Feng Wei, Shuangcheng Ma

**Affiliations:** 1National Institutes for Food and Drug Control, National Medical Products Administration, Beijing 102629, China; 2National Institutes for Food and Drug Control, Chinese Academy of Medical Sciences & Peking Union Medical College, Beijing 100050, China; 3Thermo Fisher Scientific (China) Co., Ltd., Shanghai 201206, China; 4Chinese Pharmacopoeia Commission, Beijing 100061, China

**Keywords:** heart-cutting 2D–LC, SEC–HILIC, charged aerosol detector, polysaccharides, *Dendrobium*, structural characterization

## Abstract

Natural polysaccharides are important active biomolecules. However, the analysis and structural characterization of polysaccharides are challenging tasks that often require multiple techniques and maps to reflect their structural features. This study aimed to propose a new heart-cutting two-dimensional liquid chromatography (2D–LC) method for separating and analyzing polysaccharides to explore the multidimensional information of polysaccharide structure in a single map. That is, the first-dimension liquid chromatography (^1^D–LC) presents molecular-weight information, and the second-dimension liquid chromatography (^2^D–LC) shows the fingerprints of polysaccharides. In this 2D–LC system, the size-exclusion chromatography–hydrophilic interaction chromatography (SEC–HILIC) model was established. Coupling with a charged aerosol detector (CAD) eliminated the need for the derivatization of the polysaccharide sample, allowing the whole process to be completed within 80 min. The methods were all validated in terms of precision, linearity, stability, and repeatability. The capability of the new 2D-LC method was demonstrated in determining various species of natural polysaccharides. Our experimental data demonstrated the feasibility of the whole systematic approach, opening the door for further applications in the field of natural polysaccharide analysis.

## 1. Introduction

As significantly active biological macromolecules, polysaccharides are widely distributed in animals, plants, and microorganisms. They are biopolymers composed of at least 10 monosaccharides connected by glycosidic bonds, with complex and diverse structures [1]. A large number of studies have highlighted various biological activities and functions of natural polysaccharides, making them a significant component in the development of natural medicines [1,2]. The biological activity of polysaccharides is closely related to their structural features. Therefore, strict quality control of the active polysaccharide components is necessary if they are to be further developed into commercially available drugs [3,4]. However, analyzing natural polysaccharides is challenging due to their complex and diverse chemical structures that lack ultraviolet absorption. This necessitates not only lengthy and complicated sample-preparation processes but also the use of various analytical techniques for comprehensive speculation and analysis. For example, the total polysaccharide content is determined by the colorimetric method; the composition of monosaccharides is analyzed by gas chromatography or high-performance liquid chromatography; the positions of glycosidic bonds of polysaccharides are estimated by gas chromatography–mass spectrometry (GC–MS) or nuclear magnetic resonance (NMR); and the relative molecular weights of polysaccharides are determined by high-performance size-exclusion chromatography–multi-angle light-scattering–refractive index detection (HPSEC–MALLS–RID), as well as from the fingerprints of the partial hydrolysis of acids and enzymes to obtain the “saccharide mapping”, to carry on the quantitative and qualitative analysis of polysaccharides in a comprehensive way [5,6,7,8]. Although the polysaccharide multivariate fingerprint and saccharide mapping have developed in recent years [9,10,11], they either rely on the joint analysis of multiple technologies or reflect limited polysaccharide characteristics. Thus, the analysis and structural characterization of polysaccharides are challenging. Therefore, exploring the application of novel techniques in the field of polysaccharides to characterize richer structural features of polysaccharides or reflect the specificity of polysaccharides from different plant sources is particularly important.

2D–LC, as a powerful analytical technique, can connect two independent chromatographic columns with different separation mechanisms in series to form a separation system, thus significantly increasing the peak capacity and selectivity of the system [12,13]. Moreover, 2D–LC is commonly categorized into comprehensive 2D-LC and heart-cutting 2D-LC, depending on whether all or selected portions of the one-dimensional (^1^D) eluates are separated in 2D–LC [13]. At present, 2D–LC is mostly used for separating and analyzing small molecules, which is important for certain difficult-to-separate components [14,15,16,17]. For example, conventional HPLC cannot completely separate them due to the extremely similar structure of ginsenoside compounds. Yao et al. [14] isolated and quantified the major *P. notoginseng* saponins such as Rg1 and Re, which are difficult to isolate, by 2D-LC. In current macromolecule assays, 2D–LC is mainly applied to proteins [18,19,20], and less of the literature deals with analyzing saccharide substances [21,22].

Based on the essential characteristics of polysaccharides, the column combination for the 2D–LC analysis mode in this study was SEC–HILIC. First, the molecular weight and distribution are important characteristics of polysaccharides. They have a close relationship with the activity, which is a key indicator for controlling product quality [23]. SEC is widely used in characterizing biomolecules, and the use of SEC to characterize the molecular weight of polysaccharides is well documented [24,25,26]. Meanwhile, HILIC can retain and separate polar compounds and is now widely used in the analysis of saccharides substances [27,28,29,30]. In summary, SEC (^1^D) separates the polysaccharide samples based on their molecular weight from large to small, and then selects the corresponding retention time components to enter HILIC (^2^D) for separation and analysis. In this case, the combination of SEC–HILIC to establish a novel method for analyzing polysaccharides would be a very promising alternative.

Second, in selecting the detectors for 2D-LC, most of the current coupled detectors include MS, the Diode array detector (DAD), RID, and other detectors. However, they have certain shortcomings: most polysaccharides lack ultraviolet (UV) absorption, are hard to ionize, and have a wide range of isomers, making their identification by MS more difficult [31]. DAD is difficult to detect in the absence of chromophores, whereas RID is incompatible with gradient elution and is susceptible to the interference of external factors [14,32,33]. In recent years, CAD, as a new general-purpose detector, has shown wide applicability in the field of analysis, especially for analyzing and detecting saponins, saccharides, and alkaloids with no or weak UV absorption [34,35,36]. CAD is based on the atomization–aerosol principle, which is less affected by the volume flow rate and temperature; it is suitable for gradient elution. CAD is particularly effective in analyzing and determining oligosaccharides without derivatization and is two to six times more sensitive than an evaporative light-scattering detector [37,38]. Therefore, the combination of 2D–LC with CAD is applied to analyze polysaccharides without derivatization, which is easier and more efficient.

Currently, the depolymerization of polysaccharides into oligosaccharides is a common and important strategy used for the qualitative and quantitative analysis of polysaccharides. In this study, treating polysaccharide samples using partial acid hydrolysis could simplify their complex structures and degrade them into oligosaccharides or monosaccharides with different degrees of polymerization. Then, it was combined with 2D–LC, along with a CAD detector, to realize an in-depth study on the structural analysis and quality control of polysaccharides (See Figure 1 for a summary of the ideas in this paper). First of all, we selected three plants with the same source of food and medicine, including *Dendrobium huoshanense*, *Astragalus membranaceus* (Fisch.) Ege., and *Pleuropterus multiflorus* (Thunb.) Nakai., which are all traditional Chinese medicines and can be used as nutraceuticals or tea beverages. And, it has been confirmed by our team that polysaccharides are the main ingredient contained. Then, the polysaccharide components were extracted from them, and their 2D-LC analysis was established. In this way, we aimed to study the applicability of different polysaccharide components in 2D–LC. Additionally, we also attempted to apply 2D-LC to the identification of traditional Chinese medicine (TCM). For example, we specifically selected six different species of *Dendrobium*, such as *D. huoshanense*, *D. officinale*, *D. nobile*, and so on. Then, we judged whether the newly established 2D–LC method could distinguish them effectively and reflect their exclusivity. Ultimately, this was used to comprehensively determine whether the method could be widely applied to the analysis and characterization of natural polysaccharides.

## 2. Materials and Methods

### 2.1. Chemicals and Materials

Reference substances (dextran, D1–D7) were obtained from the National Institute for Food and Drug Control (NIFDC, Beijing, China). Chromatography-grade acetonitrile and ammonium formate were obtained from Tedia Company Inc. (Fairfield, OH, USA). Ultrapure water was prepared using a Millipore Milli-Q purification system (Millipore, St. Louis, MA, USA). Other reagents were of analytical grade, except where noted. All solutions were prepared daily. The mobile phases used for HPLC were filtered (0.45 μm) and ultrasonically degassed before use.

*D. huoshanense*, *D. officinale*, *D. nobile*, *D. aphyllum*, *D. devonianum* Paxton, *D. pierardii*, *Astragalus membranaceus* (Fisch.) Ege., and *Pleuropterus multiflorus* (Thunb.) Nakai were purchased from various collection areas in China. The polysaccharides extracted from *Dendrobium* species (1–6) were named in the following order: DHP, DOP, DNP, DAP, DDP, and DPP; other polysaccharides (7–9) included AMP, PMP, and dextran. Additionally, all Chinese medicinal materials had been identified by NIFDC botanists.

### 2.2. Sample Preparation

Referring to the extraction method (water extraction and alcohol precipitation) of Dendrobium polysaccharides in Chinese pharmacopoeia [23], the herbs were first crushed into a powder and passed through a No. 3 sieve. Then, 3 g of the powder was weighed precisely, mixed with 200 mL of water, heated at 100 °C and refluxed for 2 h. The mixture was filtered and concentrated to a certain volume, and then anhydrous ethanol was added to reach the concentration of 80% to precipitate the polysaccharides. The concentrate was kept in the refrigerator at 4 °C, and, after 12 h, taken out, and centrifuged for 10 min (5000 rpm). The supernatant was discarded, and the precipitate was evaporated, then dissolved in water. Then, the proteins in the crude polysaccharides were removed by adding one-fifth of the volume of Sevage reagent (trichloromethane:n-butanol, 4:1, *v*/*v*), shaken well for 30 min, and centrifuged at 5000 rpm for 10 min. The supernatant was collected, and the process was repeated several times until no denatured protein remained at the junction. The deproteinized polysaccharide extract was repeatedly evaporated on a water bath to remove organic reagents from the polysaccharide solution to obtain crude polysaccharide.

Each sample (6 mg/mL) was hydrolyzed with trifluoroacetic acid (TFA, 1 mol/L) at 80 °C for 90 min. The sample was blow-dried under nitrogen, mixed with an appropriate amount of methanol, and blow-dried again. The process was repeated three times to remove the remaining TFA, and, further, 500 μL of water was added for complete the dissolution [39].

### 2.3. Preparation of Standard Solutions

^1^D-LC: a molecular-weight reference substance (dextran) was prepared by dissolving it in ultrapure water at a concentration of about 5 mg/mL. The relative molecular masses (molecular weight, *Mw*) of the seven controls (D1–D7) were 180, 505, 2700, 5250, 9750, 21,000, and 44,100 Da, respectively.

^2^D-LC: Dextran (*Mw* 44100) was used as a reference substance in ^2^D–LC. It was accurately weighed and added to ultrapure water to bring its concentration to approximately 12 mg/mL. Then, partial acid hydrolysis was performed following the sample treatment steps described in Section 2.2 on sample preparation.

### 2.4. Instrumentation

All experiments in this study were performed using a Vanquish 2D liquid chromatography system (Thermo Fisher Scientific, Waltham, MA, USA), the configuration of which is shown in Figure 2. The system consisted of a system base, a binary pump (acting as a ^1^D pump only), a dual pump (left pump: ^2^D pump; right pump: compensation pump), an autosampler (connected to the ^1^D pump), a CAD detector (VH-D20-A, Thermo Fisher Scientific, Waltham, MA, USA), two column compartments (each configured with a 2-Position/6-Port column switching valve and a 6-Position/7-Port column switching valve), and three viper loops. The OHpak SB-803 HQ (8.0 mm × 300 mm, 6 µm; Shodex, Yokohama, Japan) gel column was selected for ^1^D–LC analysis based on the molecular-weight range. For ^2^D–LC analysis, an HILIC chromatography column, Acclaim Trinity P2 (3.0 mm × 50 mm, 3 µm; Thermo Fisher Scientific Inc., Waltham, MA, USA), was used.

### 2.5. First-Dimension Separation

In the ^1^D separation, an OHpak SB-803 HQ column (8.0 mm × 300 mm, 6 μm; Shodex) was used with the flow rate of 1 mL/min. The mobile phase consisted of ultrapure water with an isocratic elution, and the column temperature was maintained at 40 °C. The injection volume for both the standard mixture and the sample was 10 μL. The signal evaluation was based on the peak area, and each sample or standard was injected in triplicate.

### 2.6. Two-Dimensional Liquid Chromatography

In the ^2^D separation, an Acclaim Trinity P2 column (3.0 mm × 50 mm, 3 µm; Thermo Fisher Scientific) was used with a flow rate of 0.4 mL/min. The mobile phases consisted of acetonitrile (A), water (B), and 100 mM ammonium formate (C). The compensation pump was set to a flow rate of 0.1 mL/min, and the mobile phase consisted of solvent A–acetonitrile: solvent C–ammonium formate (95:5, *v*/*v*). The specific elution conditions are depicted in Table 1. The column temperature was maintained at 40 °C, and the injection volume of the standard mixture or sample was 10 μL. The detection conditions of CAD were set as follows: data-collection rate, 5 Hz; filter constant, 3.6 s; evaporator temperature, 40 °C; and power function value, 1.0. The nitrogen pressure of CAD was adjusted to 60 psi, and the response range was set to 100 pA. The data were recorded and processed using Chromeleon 7.2 software (Thermo Fisher Scientific, Waltham, MA, USA). The 2D–LC system switched between 1-2 and 6-1 modes. The details of the gradient program used are provided in Table 2. Here, it is worth mentioning that the ^1^D–LC analysis time was 15 min, after which the flow rate of the ^1^D pump was adjusted to a lower flow rate (0.1 mL/min) into the waste stream in order to protect the one-dimensional column and save the mobile phase. Overall, three-time heart-cutting actions took place within 80 min for one chromatographic analysis. Data were processed by Chempattern 2017 and Chameleon 7.2 software.

### 2.7. Method Validation

Five batches of *D. huoshanensis* from different origins were extracted and examined as described in Section 2.2 on sample preparation and Section 2.6 on 2D–LC. Then, the obtained profile files were imported into the 2012 version of the similarity evaluation system for the chromatographic fingerprint of TCM to evaluate the reproducibility of the samples [40,41]. According to the proposed chromatographic conditions, the reference substance (dextran, *Mw* 44100) with a known concentration was subjected to an acid hydrolysis treatment and was injected six times continuously. Six peaks were selected, the peak areas were measured, and the relative standard deviation (RSD) was calculated to measure the precision. Six parallel experimental solutions were prepared to assess the reproducibility. Meanwhile, the test solutions were maintained at room temperature for 0, 2, 4, 6, 8, 12, and 24 h to analyze the stability of the solutions. The linearity was evaluated by setting six different concentration gradients of dextran control solutions, with concentrations set at 6, 8, 12, 16, 18, and 20 mg/mL. A linear regression analysis was performed with concentration as the independent variable and chromatographic peak area as the dependent variable. The regression equation was then employed for further analysis.

## 3. Results and Discussion

### 3.1. Optimization of Sample-Preparation Procedures

The sequential degradation of a polysaccharide structure can be realized by controlling the acid concentration, hydrolysis temperature, and time. The less stable branched chains and ends are generally hydrolyzed first during partial acid hydrolysis. This study explored the conditions for the acid hydrolysis of polysaccharide samples: (1) temperature: 60 °C, 80 °C, and 100 °C; (2) time: 60, 90, and 120 min; and (3) TFA concentration: 0.5, 1, 2, and 4 mol/mL. After the experimental analysis, if the temperature was too low, the process would take longer, and when the temperature was too high or the acid concentration was too large, the hydrolysis tended to be violent. On the contrary, when the acid concentration was low, the reaction tended to be mild, which was conducive to the hydrolysis of polysaccharides. Ultimately, the optimal acid hydrolysis conditions were selected as follows: 80 °C, 90 min, and TFA (1 mol/mL).

### 3.2. ^1^D Separation

#### 3.2.1. ^1^D Optimization

The ^1^D column used a water-soluble gel column with polymer-matrix packing, which was suitable for separating various water-soluble macromolecules, proteins, and oligomers as well as for determining molecular-weight distribution. The samples and dextran controls involved in this study were predetermined for the molecular-weight range of 1.0 × 10^3^ to 2.6 × 10^5^ Da. Therefore, the most suitable gel chromatography column, OHpak SB-803 HQ, was selected for a more efficient separation.

Further, the ^1^D separation conditions, such as the mobile phase, flow rate, and column temperature, were explored using an OHpak SB-803 HQ column. A comparative analysis revealed that the retention time was shortened when the flow rate was increased to 1.0 mL/min; however, the separation was still well maintained. In addition, given the compatibility of the ^2^D mobile phases and the CAD detector’s requirement for salt concentration, the mobile phase was mostly analyzed in 100% ultrapure water. Although higher temperatures could shorten and improve the separation better, too high temperatures could lead to lower column efficiency and reduced separation. Therefore, based on the overall consideration of the 2D system, the final liquid-phase conditions were determined as follows: flow rate, 1.0 mL/min; column temperature, 40 °C; mobile phase, 100% ultrapure water; injection volume, 10 μL; and time, 15 min.

#### 3.2.2. ^2^D Optimization

For the ^2^D analysis, the first-dimension experimental conditions were the final conditions as determined in Section 3.2.1 on the choice of ^1^D liquid-phase conditions. Acclaim Trinity P2 (3.0 × 50 mm^2^, 3 µm; Thermo Fisher Scientific) was selected as the second-dimension column. This unique high-efficiency silica-based column provided HILIC interactions besides anion- and cation-exchange properties. Adequate retention and separation could be achieved with Acclaim Trinity P2, and the analysis time was subsequently reduced because of the shorter column length, with the analysis time required for a single cut being 25 min.

First, the mobile phase of solvent A, acetonitrile, and solvent B, water, was selected for gradient elution, and the proportional changes of different mobile phases (acetonitrile concentration: 90–60%, 80–50%, and 80–40%) were analyzed sequentially. According to the nature of HILIC columns [27,28], ionic additives such as ammonium formate to the mobile phase could be used to control the pH and ionic strength of the mobile phase, which affected the polarity of the analytes and led to differential retention changes. The addition of a certain percentage of 5% ammonium formate solution (100 mM, pH 3.65) to the mobile phase improved the separation of chromatographic peaks. Similarly, as a compensation flow path, the organic phase (95% acetonitrile) was added to maintain the stability of the 2D–LC system. Due to the different polymerization degrees of the components in the three-time heart-cutting, the length of each analysis period needs to be adjusted according to the actual analysis situation. The mobile phase for each heart-cutting analysis was equilibrated from 80% acetonitrile to 50% acetonitrile and back to 80% acetonitrile for about 10 min. Among them, the third heart-cutting fraction has a lower degree of polymerization and can be better analyzed without equilibration. The column temperature at 40 °C separated better than at other temperatures (30 °C). Since the components collected from the ^2^D for further analysis have undergone some degree of dilution, the peak signal is significantly reduced in the ^2^D. Given that the mobile phase in the ^2^D to the ^1^D is excessive from the aqueous phase to the organic phase. Hence, adding a certain percentage of acetonitrile to the auxiliary pump can result in an increased signal response in ^2^D–LC. Final compensation pump conditions: the acetonitrile compensation ratio was controlled at 5% and the flow rate was 0.1 mL/min.

### 3.3. Precision, Linearity, Stability, and Repeatability

The 2D analysis results of five batches of *D. huoshanensis* polysaccharides with different origins were matched using the similarity evaluation system for chromatographic fingerprint of TCM, and the similarities were calculated to be 0.915−0.994. This indicated that the consistency between the samples was good, proving that the method had good reproducibility. Table 3 presents the precision data and other data for the 2D analytes from the working solution. The RSD values for the precision data were less than 2%. Meanwhile, the RSD values for the analytes in the repeatability assay were <2%. The RSD values of the analytical results of the working solutions stored at room temperature for 0, 2, 4, 6, 8, 12, and 24 h were less than 2%, indicating the good stability of the method. The peak areas at six concentration levels were used as a linear standard curve. In this study, good linearity was observed for each analyte (with the coefficient of determination [R^2^] > 0.999 for each analyte). These results confirmed that the proposed method had good precision, linearity, repeatability, and stability.

### 3.4. ^1^D Analysis Results

First, some polysaccharides would be degraded to oligosaccharides or monosaccharides with different degrees of polymerization after partial acid hydrolysis. Therefore, the molecular-weight control series (D1–D7) was selected, with molecular weights covering hundreds to tens of thousands. The results of the molecular-weight localization of the controls are illustrated in Figure 3A. The cubic standard curve equation was fitted with the *LgMw* of the control as the vertical coordinate and the retention time T/min as the horizontal coordinate (Figure 3B), which yielded *LgMw* = 0.0377*T*
^3^ − 0.9323*T* ^2^ + 6.8949*T* − 11.0345 (R^2^ = 0.9988), and the linear ranges of the control were in accordance with the experimental requirements. Upon comparison of the standards (D1–D7), it was found that the molecular-weight distribution of all treated samples was mainly in the range of a few hundreds to tens of thousands (Table 4), which was consistent with previous speculations.

It was hypothesized based on the standard curve equation that the oligosaccharides might be distributed between 8.7 min and 9.7 min. However, the retention time of the monosaccharides was 10.4 min, but some samples still showed peaks in some of the samples after that period. This did not exclude the possibility that the extracted polysaccharides contained other small-molecule impurities. At the same time, this was good proof that the selection of SEC as a ^1^D system served the purpose of separating other impurities based on molecular weight. Moreover, some unhydrolyzed polysaccharides could remain after the partial acid hydrolysis treatment of the samples, so the peaks also appeared before 8.7 min.

Subsequently, the molecular weight and distribution of these nine different polysaccharides were analyzed, and the molecular weights of the samples were calculated based on calibration curves obtained from a series of dextran standards to compare the molecular-weight differences before and after sample treatment. Significant differences were found in the molecular-weight distributions of samples by combining ^1^D-LC mapping (Figure 4A) and molecular-weight data, which might be related to the different actual polysaccharide contents and structures of different varieties of TCM. Meanwhile, under the same controlled-degradation conditions, the molecular weight of the sample decreased compared with that earlier. The degree of molecular-weight reduction varied among different samples, with a reduction range of approximately one to two orders of magnitude. This difference stemmed from the quality of the TCM itself, which was expected to be beneficial for further analysis in 2D-LC.

### 3.5. ^2^D Analysis Results

Oligosaccharides are low-molecular-weight polymers with 2−10 monosaccharides linked by glycosidic bonds [34]. The partial acid hydrolysis of polysaccharides to oligosaccharides decreases the molecular weight. Therefore, the oligosaccharides were first eluted and aggregated using SEC chromatography and then separated from the samples using the heart-cutting method. Next, they were transferred to HILIC (^2^D-LC) for analysis to obtain the oligosaccharide information profile. The analysis time of the first dimension was 15 min in the 2D–LC system constructed in this experiment. The retention time of the molecular-weight distribution of the ^1^D-LC chromatogram suggested that the oligosaccharides might be distributed in the range of 8.7–9.7 min. Ultimately, three consecutive heart-cuttings were performed on the sample fractions eluted from the ^1^D–LC based on the loop specification and the retention-time distribution of oligosaccharides, reducing the complexity of sample analysis. Among these, the heart-cutting time was selected to be t_1_ (8.8–9.0 min), t_2_ (9.1–9.3 min), and t_3_ (9.5–9.7 min). The results of the ^1^D–LC and ^2^D–LC (t_1_,t_2_,t_3_) analysis of polysaccharide samples are plotted in Figure 4.

^2^D–LC was eluted sequentially according to polarity from small to large. Each heart-cutting was performed under optimized conditions, facilitating the further separation of the samples. This study compared the separation ability of ^1^D–LC (SEC) and 2D–LC (SEC–HILIC) using polysaccharides as samples, with a total analysis time of 68 min for ^2^D–LC. First, combined with the reference substance (dextran, *Mw* 44100), each sample showed an oligosaccharide profile at t_1_ and t_2_. That is, elution occurred sequentially according to the degree of polymerization and presented a continuous wavy peak shape at retention times of approximately 25–30 and 50–55 min (marker points a and b), indicating the presence of oligosaccharides at time t_1_ and t_2_. However, only about three independent chromatographic peaks (marker e) were observed in the t_3_ region. Based on the separation results of ^1^D–LC (SEC), it could not be excluded that the t_3_ region also contained impurities of small-molecule compounds in the crude polysaccharides. The reason for retaining this section for heart-cutting analysis was to make a clear comparison with the t_1_ and t_2_ regions so as to support the feasibility of this experimental approach. When integrating data from all three time periods, the heart-cutting analysis after each ^2^D–LC separation revealed more than one chromatographic peak, with up to nine times as many peaks as in ^1^D–LC. This indicated that the resolution of oligosaccharide fractions eluted by molecular weight was significantly improved after further separation based on different polarities. Meanwhile, 2D-LC was demonstrated to have a higher peak capacity and separation selectivity, and this analytical method was suitable for separating polysaccharides.

Subsequently, four different sources of polysaccharides were analyzed: *D. huoshanense*, *Astragalus membranaceus* (Fisch.) Bunge, *Polygonum multiflorum* Thunb, and dextran (1. DHP, 7. AMP, 8. PMP, and 9. dextran). Although polysaccharides from different sources had oligosaccharide characteristic peaks at a and b, still significant differences existed. Dextran was used as a reference substance hydrolyzed to oligodextrose, with a single component, and the map showed results as expected. Compared with the reference substance dextran, AMP and PMP also exhibited clear oligosaccharide characteristic chromatographic peaks at positions a and b (Figure 4), indicating that these two TCMs contained glucan. However, significant differences were found between DHP and other TCM polysaccharides in the region. We preliminarily speculated that this was a heteropolysaccharide chain that could not be separated well. Furthermore, existing research indicated that the polysaccharides present in DHP were primarily glucomannans [42]. In addition, significant differences were observed in the c, d, and e regions. Particular attention should be paid to the fact that PMP appeared to have characteristic peaks at f, g, and h, which were clearly different from those of other polysaccharides. In summary, the applicability of the presently established 2D–LC method in different varieties of polysaccharide compositions was preliminarily demonstrated.

Then, we also attempted to apply 2D–LC to identify TCM, distinguishing six different types of *Dendrobium* polysaccharides. As shown in Figure 4B, all *Dendrobium* samples exhibited oligosaccharide characteristics in both a and b. Combined with the 2D–LC fingerprints of polysaccharides from different sources in Figure 5A, it was observed that DNP, DAP, DDP, and DPP exhibited obvious common peaks at positions c, d, and e, and DHP and DOP were significantly different from the other *Dendrobium* polysaccharides. Therefore, nine different sources of polysaccharides were imported into ChemPattern software for similarity analysis. The similarity data in Figure 5B showed that DOP was indeed more similar to DHP, which could reach about 0.890. In addition, the similarity values of the other *Dendrobium* varied widely, ranging from 0.600 to 0.800. *D. huoshanense* and *D. officinale*, as the mainstream *Dendrobium* species, have the highest market prices of all *Dendrobiums*, and both have similar quality-control indicators. These findings indicated that, although the polysaccharide profiles of different species of *Dendrobium* had similar ^1^D-LC, significant differences were noted in ^2^D-LC. Meanwhile, the proposed method could also better distinguish different varieties of TCM polysaccharides, as shown by the analysis of the similarity evaluation of AMP, PMP, and dextran. In conclusion, this method may assist in the quality-control studies of *Dendrobiums*, which can distinguish between different *Dendrobium* spp. and can also be used as a basis for identifying TCMs.

## 4. Conclusions

This structural heterogeneity of polysaccharides impairs their comprehensive characterization and requires analytical techniques with high resolving power and sensitivity. In the qualitative and quantitative analysis of polysaccharides, the degradation of polysaccharides into oligosaccharides is a commonly used key method. For instance, Li et al. [3,43,44] performed partial acid hydrolysis on polysaccharides followed by derivatization, enabling a comprehensive characterization of various plant polysaccharides using saccharide-mapping techniques such as HPTLC and PACE. Although this method is innovative, it demands extensive analytical technology and a substantial workload. Due to the unique principle of CAD, the analysis of carbohydrate compounds can be accomplished without the need for derivatization, saving considerable time. The oligosaccharide profiles obtained from different natural polysaccharides under the same controlled degradation conditions exhibit high variability and stability. Hence, utilizing the polysaccharide-controlled degradation of oligosaccharides for the identification of traditional Chinese medicines (TCMs) is a rational strategy.

2D–LC, as an advanced separation technique, is relatively uncommon in the analysis and separation of plant polysaccharides. The primary reason for this phenomenon may lie in the complex and heterogeneous structures of polysaccharides, which pose significant challenges to the development of innovative characterization techniques tailored to their properties. This study successfully designed and independently established a 2D–SEC–HILIC system, which proved efficient for the separation of plant polysaccharides. The system showed an excellent separation ability when analyzing complex polysaccharide samples. In 2D–LC, the application of not only ^1^D (SEC) for characterizing the molecular weight and distribution of polysaccharide samples but also using its separated components to further resolve into multiple peaks in ^2^D (HILIC) indicated a significant complementarity between the two separation modes. This combination is favorable for the 2D separation of samples and is suitable for constructing a 2D separation system. However, certain areas still need improvement in this experiment, such as the subsequent use of polysaccharides to purify the samples and reduce the interference of small molecules, and the possibility of connecting mass spectrometry for identifying information. This will be a part of our next experimental research program.

In conclusion, the 2D–LC method established in this study helped achieve far better separation effects and information than the conventional HPLC method, which proved the feasibility of the method established. Through these research and optimization efforts, 2D–LC is expected to become a powerful tool for the separation and analysis of plant polysaccharides and provide new perspectives for revealing the fine structural characterization of polysaccharides.

## Figures and Tables

**Figure 1 foods-13-01173-f001:**
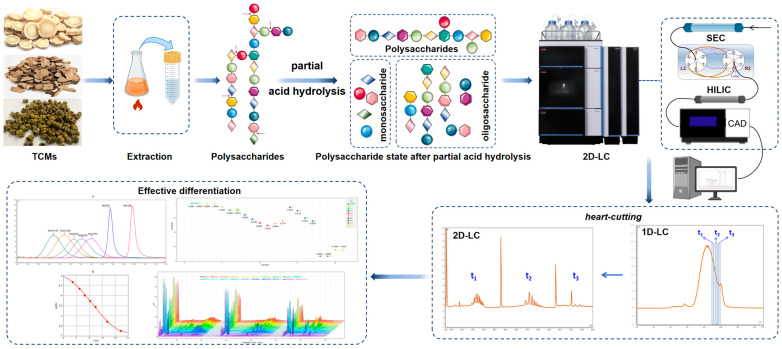
Graphical summary of natural polysaccharides analyzed by 2D-LC.

**Figure 2 foods-13-01173-f002:**
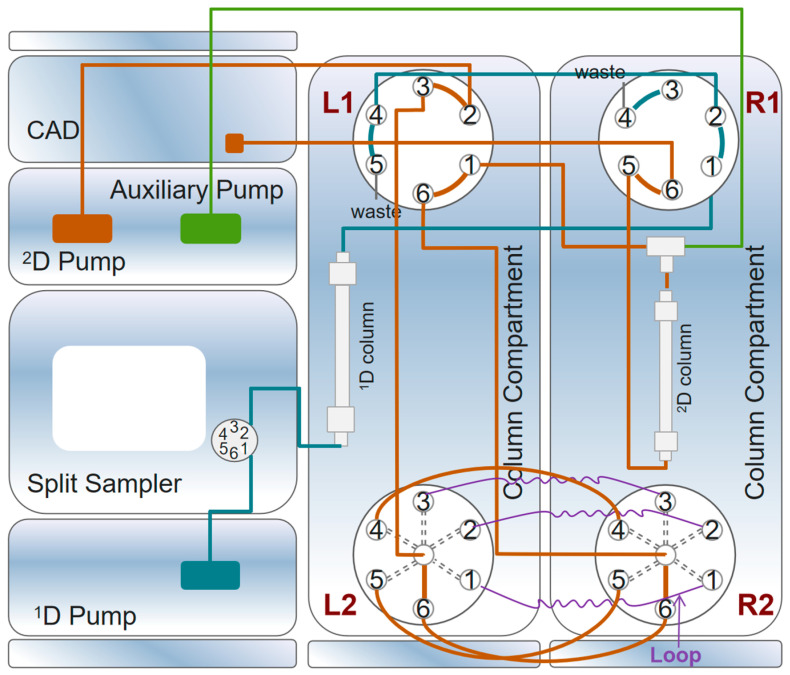
Online heart-cutting 2D-LC system connection.

**Figure 3 foods-13-01173-f003:**
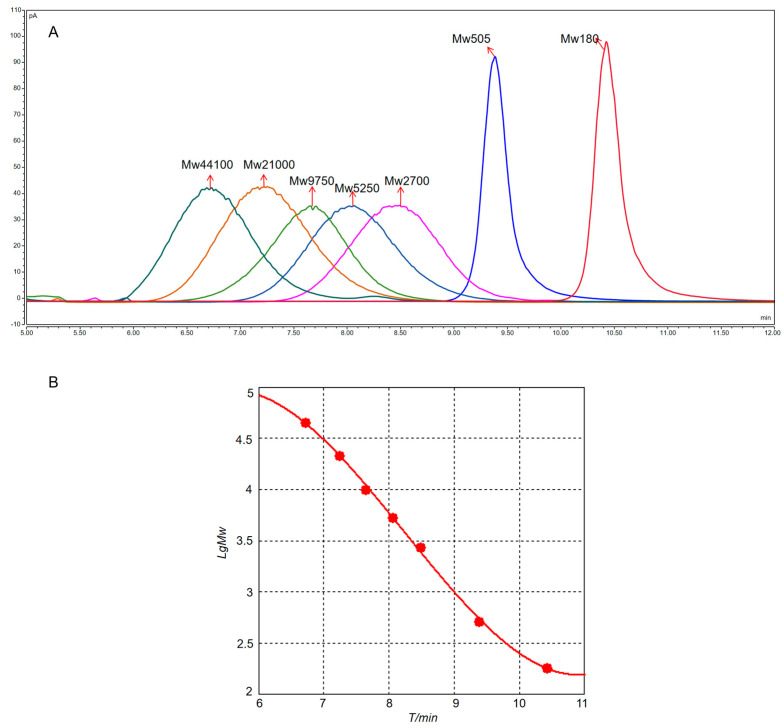
(**A**) Liquid chromatogram of D1–D7 molecular-weight standards. (**B**) Standard fitting curve of molecular weight.

**Figure 4 foods-13-01173-f004:**
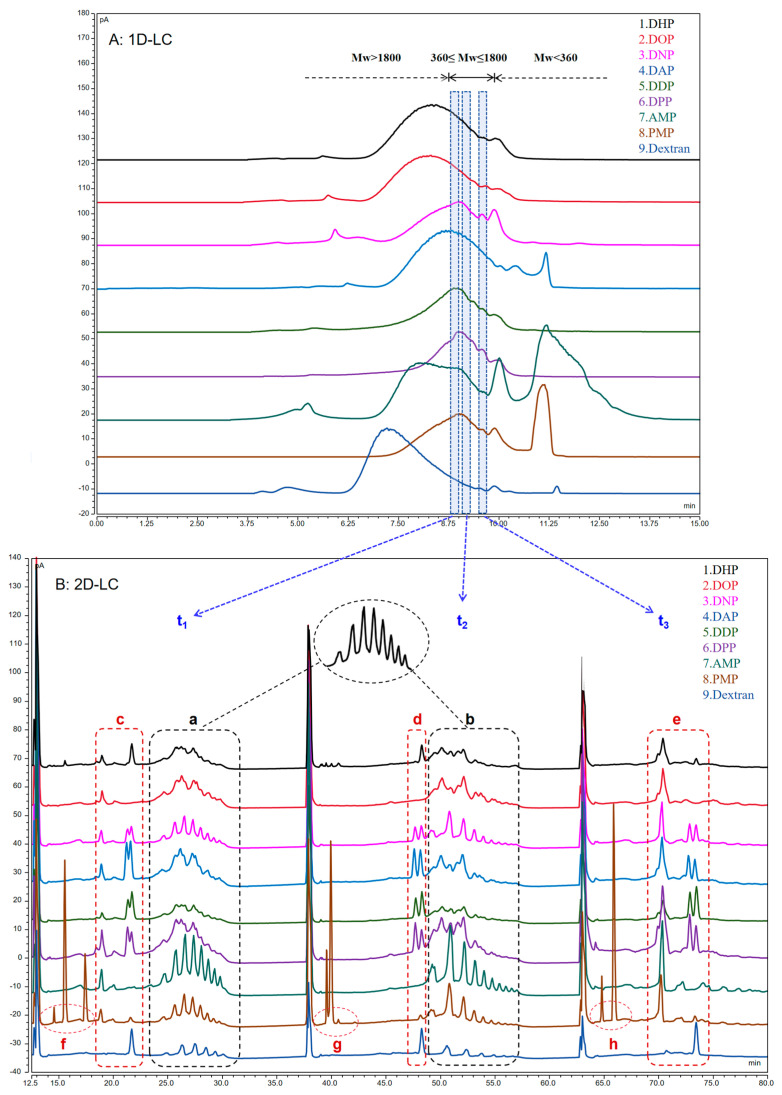
Chromatograms in (**A**) ^1^D-LC and (**B**) ^2^D-LC (different polysaccharides, in order: 1. DHP, 2. DOP, 3. DNP, 4. DAP, 5. DDP, 6. CPP; 7. AMP, 8. PMP, and 9. dextran).

**Figure 5 foods-13-01173-f005:**
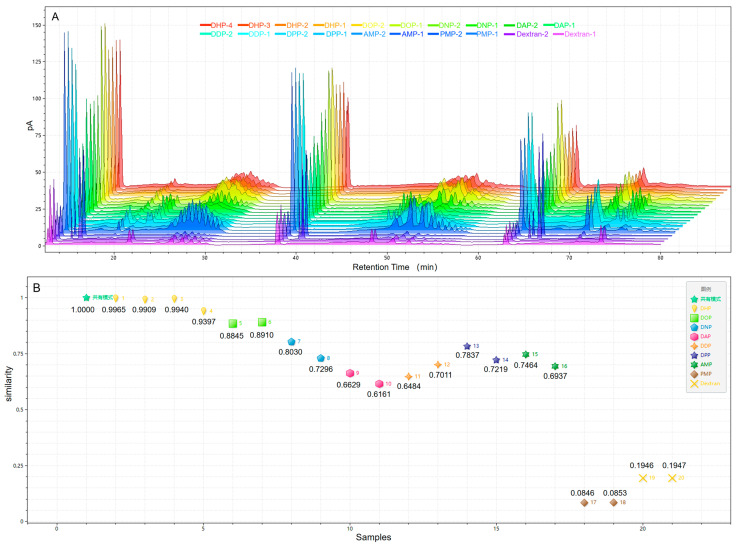
(**A**) Fingerprints of different natural polysaccharides; (**B**) evaluation of similarity.

**Table 1 foods-13-01173-t001:** Optimization of gradient elution procedure for 2D-LC determination method.

^1^D Separation (^1^D Pump)				^2^D Separation (Left Pump)			
Time (min)	Flow (mL/min)	B (%) H_2_O		Time (min)	Flow (mL/min)	A(%) CH_3_CN	B(%) H_2_O
0	1.0	100		0.00	0.400	80	20
15.00	1.0	100		12.00	0.400	80	20
15.10	0.1	100		27.90	0.400	50	50
80.00	0.1	100		28.00	0.400	80	20
**Auxiliary Pump (Right Pump)**				40.00	0.400	80	20
Time (min)	Flow (mL/min)	A (%) CH_3_CN	C (%) NH_4_COOH	54.90	0.400	50	50
				55.00	0.400	80	20
0	0.1	95	5	65.00	0.400	80	20
80.00	0.1	95	5	80.00	0.400	50	50

**Table 2 foods-13-01173-t002:** Description of valve switching.

Time (min)	Valve-Switching Position		Valve-Position Description
	L1 L2	R1 R2	
0	1_2 6	6_1 6	Start ^1^D analysis; re-equilibrate ^2^D chromatographic column
8.80	1_2 1	1_2 1	Loop 1 is collecting the substance
9.00	1_2 1	1_2 1	
9.01	1_2 6	6_1 6	Loop 1 terminates the collection
9.10	1_2 2	1_2 2	Loop 2 is collecting the substance
9.30	1_2 2	1_2 2	
9.31	1_2 6	6_1 6	Loop 2 terminates the collection
9.50	1_2 3	1_2 3	Loop 3 is collecting the substance
9.70	1_2 3	1_2 3	
9.71	1_2 6	6_1 6	Loop 3 terminates the collection
12.00	6_1 1	1_2 1	Starting ^2^D analysis of substance in Ring 1
37.00	6_1 2	1_2 2	Starting ^2^D analysis of substance in Ring 2
62.00	6_1 3	1_2 3	Starting ^2^D analysis of substance in Ring 3
80.00	6_1 3	1_2 3	Termination analysis

**Table 3 foods-13-01173-t003:** Linearity, precision, repeatability, and stability of the established 2D method.

Retention Time	Linearity (Regression Equation)	R^2^	Precision (*n* = 6)	Repeatability (*n* = 6)	Stability (*n* = 6)
			RSD (%)	RSD (%)	RSD (%)
26.697	Y = 0.1395X + 0.3452	0.9990	1.78%	1.76%	1.76%
27.930	Y = 0.1414X + 0.4475	0.9990	1.15%	1.27%	1.75%
29.000	Y = 0.0955X + 0.2788	0.9990	1.30%	1.48%	1.83%
50.900	Y = 0.1203X + 0.1517	0.9996	1.06%	1.47%	1.75%
52.730	Y = 0.0659X + 0.0282	0.9993	1.34%	1.86%	1.85%
54.177	Y = 0.0234X + 0.0495	0.9990	1.34%	1.68%	1.81%

**Table 4 foods-13-01173-t004:** Comparison of the molecular weights of samples before and after partial acid hydrolysis (^1^D–LC).

Polysaccharide Samples	Stock Solution (*Mw*/Da) (n = 2)	Solution after Partial Acid Hydrolysis (*Mw*/Da) (n = 2)
1. DHP	1.426 × 10^5^	9.775 × 10^3^
2. DOP	1.998 × 10^5^	7.854 × 10^3^
3. DNP	1.774 × 10^5^	1.141 × 10^4^
4. DAP	1.932 × 10^5^	7.445 × 10^3^
5. DDP	1.891 × 10^5^	7.839 × 10^3^
6. DPP	2.270 × 10^5^	6.636 × 10^3^
7. AMP	8.098 × 10^4^	1.049 × 10^4^
8. PMP	2.560 × 10^5^	4.781 × 10^3^
9. Dextran	4.410 × 10^4^	2.273 × 10^4^

## Data Availability

The data are contained within the article.

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
