# Peer review of "An Analysis of Polysaccharides from Eight Plants by a Novel Heart-Cutting Two-Dimensional Liquid Chromatography Method"

_foods, 2024, doi:10.3390/foods13081173_

Round 1

Reviewer 1 Report

Comments and Suggestions for Authors

1. Title: Please revise to read: Analysis of polysaccharides from six plants by a novel heart-cutting two-dimensional liquid chromatography method. Heart-cutting is not an uncommon word in Analytical Chemistry, and should not for any reason be put in parenthesis.

2. In the abstract and main text, care should be taken with writing 2D-LC and 1D-LC. The numbers are not superscripts!

3. In L18, SEC – HILIC should be expanded at first use.

4. Words already used in the title are not useful as indexing keywords. Please use other terms.

5. L30-32: Please add a citation.

 -L40: technical means >> analytical techniques.

-Please avoid the use of etc in the manuscript. Also define abbreviations whenever they are used at the first time in the main text.

L61: small-molecule compounds >> small molecules.

The DISCUSSION section needs to be expanded, specifically referring to previous studies.

Other suggestions are in the attached PDF file.

Comments on the Quality of English Language

The manuscript could benefit from English language proofreading by a proficient speaker.

Reviewer 2 Report

Comments and Suggestions for Authors

Review comments on

 The analysis of natural polysaccharides by a new “heart-cutting” two-dimensional liquid chromatography method

 The study aimed to propose a new "heart-cutting" two-dimensional liquid chromatography (2D-LC) method for separating and analyzing polysaccharides to explore the multidimensional information of polysaccharide structure. A question concerning the aim and title is if the heart catting is a new method?   

The manuscript is well written, Introduction in informatory and  methods are well described. The discussion is relevant to the obtained study results. The applicability of 2D-LC method in different varieties of polysaccharide compositions was preliminarily demonstrated.

The study provides a lot of technical and difficult information about the procedures. Please try to underline the most important in conclusion section, which is missing at the moment.

The manuscript is also carefully edited. Some mistakes or errors ere underlined bellow:

Line 66, 67, 88 and so on :

Line 151, 186: .[39], space and similar

Line 202: Title repeated

Table 3. “This is a table. Tables should be placed in the main text near to the first time they are cited”. Correct the title of the table

Table 3. Please provide more clear information how the parameters (precision, stability, and repeatability) are calculated or what they express. More, note the values. It concerns all the three. For example: Precision 1,78% seems, at the first sight, to be very low. It is natural to expect rather 98%

In summary, the paper resents interesting information on the procedures of polysaccharides mapping. It can be published in after Foods after very minor revision.

PS. I have notted that a similar (the same) paper was published online by the Authors under the Elsevier Creative Commons licensing. However, I am not able to recognize the potential conflict of interest or copyright violation. 

The Analysis of Natural Polysaccharides by a New “Heart-Cutting”Two-Dimensional Liquid Chromatography Method

Number of pages: 27 Posted: 14 Dec 2023

Round 2

Reviewer 1 Report

Comments and Suggestions for Authors

All comments have been addressed